# Mobilome and Resistome Reconstruction from Genomes Belonging to Members of the *Bifidobacterium* Genus

**DOI:** 10.3390/microorganisms7120638

**Published:** 2019-12-02

**Authors:** Walter Mancino, Gabriele Andrea Lugli, Douwe van Sinderen, Marco Ventura, Francesca Turroni

**Affiliations:** 1Laboratory of Probiogenomics, Department of Chemistry, Life Sciences and Environmental Sustainability, University of Parma, 43124 Parma, Italy; walter.mancino@studenti.unipr.it (W.M.); gabrieleandrea.lugli@unipr.it (G.A.L.); marco.ventura@unipr.it (M.V.); 2School of Microbiology, APC Microbiome Institute, University College Cork, Cork T12 K8AF, Ireland; d.vansinderen@ucc.ie; 3Microbiome Research Hub, University of Parma, 43124 Parma, Italy

**Keywords:** bifidobacteria, genomics, mobile elements, antibiotic resistance genes

## Abstract

Specific members of the genus *Bifidobacterium* are among the first colonizers of the human/animal gut, where they act as important intestinal commensals associated with host health. As part of the gut microbiota, bifidobacteria may be exposed to antibiotics, used in particular for intrapartum prophylaxis, especially to prevent *Streptococcus* infections, or in the very early stages of life after the birth. In the current study, we reconstructed the in silico resistome of the *Bifidobacterium* genus, analyzing a database composed of 625 bifidobacterial genomes, including partial assembled strains with less than 100 genomic sequences. Furthermore, we screened bifidobacterial genomes for mobile genetic elements, such as transposases and prophage-like elements, in order to investigate the correlation between the bifido-mobilome and the bifido-resistome, also identifying genetic insertion hotspots that appear to be prone to horizontal gene transfer (HGT) events. These insertion hotspots were shown to be widely distributed among analyzed bifidobacterial genomes, and suggest the acquisition of antibiotic resistance genes through HGT events. These data were further corroborated by growth experiments directed to evaluate bacitracin A resistance in *Bifidobacterium* spp., a property that was predicted by in silico analyses to be part of the HGT-acquired resistome.

## 1. Introduction

Bifidobacteria are gram-positive, anaerobic, non-motile, and non-spore-forming bacteria with a high G + C genomic content [1]. They represent one of the dominant microbial groups inhabiting the gastrointestinal tract (GIT) of humans and animals, including mammals, birds, and social insects [2,3,4]. Members of the *Bifidobacterium* genus are believed to be crucial for the development of a healthy gut microbiota in early life, colonizing the GIT within the first days following birth [5]. Notably, exposure to antibiotic agents as a result of intrapartum prophylaxis, commonly applied to prevent group B *Streptococcus* infections [6,7] and in infants, presenting a highly dynamic microbiota [8], may disrupt the balance between microbial members of the gut microbiota [9,10,11]. It has previously been demonstrated that the relative abundance of bifidobacterial species in the gut microbiota of infants who had not been exposed to any antibiotic treatment is higher than that in children that had been subjected to antibiotic therapy [12,13]. The extensive use of antibiotics can promote the development of antibiotic resistance in members of the microbiota and consequently in the selection of antibiotic-resistant microorganisms [11]. In this context, the collective genetic arsenal responsible for conferring antibiotic resistance (AR) through inactivation and/or removal of antibiotics is commonly referred to as the resistome [14,15,16]. The occurrence of *AR* genes may increase the ecological fitness of a bacterium and thus its ability to colonize and persist in a specific environment [17]. Recent studies have indicated frequent occurrence of horizontal gene transfer (HGT) events among bacteria residing in the gut of humans and animals [18,19]. The presence of *AR* genes in mobile genetic elements (MGEs) or near transposable elements, in pathogenic and non-pathogenic microorganisms, may be the cause for the relatively frequent transfer of such elements to human/animal pathogens or to other non-resident microorganisms of the gastrointestinal tract [19,20].

The mobilome of a microorganism refers to the collection of all MGEs, including transposases, insertion elements (IS elements), and also plasmids and prophages [21,22,23]. A recent study investigating the presence of prophages in 48 members of the genus *Bifidobacterium* predicted the presence of 90 different prophages, called bifidoprophages [24].

In the current study, we reconstructed the entire mobilome of the genus *Bifidobacterium*, including prophages and transposases, based on 625 different bifidobacterial genomes belonging to 67 different (sub)species [25]. Moreover, we reconstructed the bifidobacterial resistome, i.e., the genes whose products are predicted to be responsible for resistance against antibiotic molecules. Combining the gathered bifido-mobilome with the bifido-resistome data, we identified genetic insertion signatures that may be involved in horizontal transfer of *AR* genes in bifidobacterial genomes.

## 2. Materials and Methods

### 2.1. Bacterial Strains

We retrieved the genome sequence of 625 public available *Bifidobacterium* genomes from the National Center for Biotechnology Information (NCBI) public database (Appendix A). Collected genomes with more than 100 genomic sequences were discarded to analyze high quality genome sequences only. As reported in Appendix A, strains that were not classified at the species level were validated using the average nucleotide identity (ANI) approach. Strains used for this analysis were compared with the 67 type strains of the *Bifidobacterium* genus. Notably, two bifidobacterial strains displaying an ANI value of <94% may be considered to belong to two different species [25,26,27,28].

### 2.2. IS Elements Identification

Predicted genes of 625 bifidobacterial strains used for this study were screened for the presence of IS elements. We used a custom database composed of 329,372 RefSeq sequences belonging to the Actinobacteria phylum, retrieved from the NCBI database. The alignment was performed through BLASTP analysis with an E-value cutoff of 1e^−5^ [29]. After the manual control of the sequences with an amino acid length less than 100 amino acids of the type strains of the species studied, we decided to discard these sequences because they were considered non-functional or truncated. Finally, the selected IS element sequences were validated and classified into IS families using the IS finder database [30].

### 2.3. Bifidoprophages Identification

The 625 bifidobacterial genomes were screened for prophage-like elements using a custom database based on already identified sequences through BLASTP analysis [29] (E-value cutoff of 1e^−5^). The custom database was constructed through previously bifidoprophage-validated sequences retrieved from 60 bifidoprophages identified by Lugli et al. [24], considering genetic islands presenting different genes encoding for phage functions. Following this, a manual examination of the DNA region surrounding a putative phage-encoding gene was performed. These manual screenings allowed us to identify complete prophage-like sequences while discarding incomplete or remnant phage sequences, as previously performed by Lugli et al. [24].

### 2.4. Prediction of the Antibiotic Resistance Genes

The in silico proteome of the 625 *Bifidobacterium* genomes used in this study was screened for proteins that can act as antibiotic resistance proteins through inactivation and/or removal of antibiotic molecules. The screening was carried out using the MEGAREs database through BLASTP analysis (E-value cutoff of 1e^−18^) [29,31]. The E-value cutoff was chosen based on a manual editing performed to identify false positive sequences. The core database was obtained by non-redundant compilation of sequences contained in Resfinder, ARG-ANNOT, the Comprehensive Antibiotic Resistance Database (CARD), and the NCBI Lahey Clinic beta-lactamase archive [32,33,34,35]. Following this, a manual examination of the sequence with an E-value less than 1e^−18^ was performed in order to explore all the biodiversity of the *AR* genes of the *Bifidobacterium* species. We excluded the putative *AR* genes encoding for transporters for low accuracy in their prediction [16]. The predicted *AR* genes were classified according to the presumed mechanism of action and the antibiotic molecules they counteract.

Moreover, for the 625 bifidobacterial genomes analyzed, we manually evaluated the genes flanked by the predicted *AR* genes, forming the *Bifidobacterium* resistome, in order to identify mobile genetic hotspots that may promote HGT events.

### 2.5. Phylogenomic Analyses

The nucleotide similarity of each obtained bifidoprophage sequence was calculated using the software package LAST [36]. Results were employed to build a matrix representing the genome similarity among different prophage and to generate a clustering tree. The bifidophage sequences were aligned using Mafft software [37] and the clustering tree was constructed using ClustalW [38]. The constructed clustering tree was visualized using the FIGTREE software (http://tree.bio.ed.ac.uk/software/figtree/).

### 2.6. Bacitracin A Antibiotic Susceptibility Tests

The minimal inhibitory concentration (MIC) breakpoints (micrograms per milliliter) of bacitracin A were determined using the broth microdilution method (MDIL) according to the ISO standard guidelines [39]. Bacitracin A antibiotic was purchased from Merck (Germany). Microplates were incubated under anaerobic conditions for 48 h at 37 °C. Cell density was monitored by optical density measurements at 600 nm (OD600) using a plate reader (BioTek, Vermont, USA). The MIC breakpoint represents the highest concentration of a given antibiotic to which a particular bacterial strain is resistant.

### 2.7. Statistical Analyses

SPSS software (IBM, Italy) was used to perform statistical analysis between the BacA strains group and control group by T-student test.

## 3. Results and Discussion

### 3.1. The Putative Resistome of the Genus Bifidobacterium

In order to investigate the genetic AR arsenal carried by members of the *Bifidobacterium* genus, we investigated the resistome of 625 bifidobacterial genomes. We enlarged the previously published database on the resistome of the *Bifidobacterium* genus, which were based on 91 different genomes [16]. Putative *AR* genes encoding transporters were excluded from this analysis due to the inaccuracy of their bioinformatic prediction [16]. The overall number of putative antibiotic-resistance genes identified among these 625 genomes was 13,870, representing less than 1% of the total *Bifidobacterium* genes analyzed (Appendix A). According to the predicted mechanism of action and the antibiotic molecules that could be counteracted, seven different AR gene classes were identified (Figure 1). The AR class with the highest number of representatives was the one conferring glycopeptide resistance, which corresponds to 5999 putative enzymes acting against glycopeptide antibiotics, such as vancomycin, teicoplanin, and telavancin (Figure 1) [40,41,42]. Notably, *Bifidobacterium bifidum* 791, *Bifidobacterium longum* subsp. *infantis* 1888B and *B. bifidum* AM42-15AC were strains containing the highest number of genes predicted to belong to this glycopeptide-resistance class, each encoding 29 distinct enzymes predicted to confer such resistance. Moreover, we identified 2178 genes putatively belonging to the tetracycline-resistance class (Figure 1) [43,44,45,46]. Members of the *B. bifidum* species, isolated from fecal samples of healthy Chinese individuals [47], i.e., strains TM05-15, TF07-22, TM02-15, TM02-17, TM06-10, and TM07-4AC, were shown to contain the highest number of genes encoding proteins predicted to counteract tetracycline antibiotics, ranging from 29 genes of *B. bifidum* TM05-15 to 28 genes for the other *B. bifidum* strains. Notably, 484 analyzed strains did not appear to contain genes encoding tetracycline-resistance proteins, representing 77.5% of the total *Bifidobacterium* strains analyzed. Furthermore, 2437 genes were found to belong to the beta-lactamase class and *Bifidobacterium animalis* subsp. *animalis* ATCC 25527 was shown to be the strain with the highest number (i.e., 32) of predicted beta-lactamase-encoding genes, while 469 of the 625 analyzed genomes did not appear to encompass genes belonging to this AR class (Figure 1).

Moreover, 2618 genes were predicted to belong to the methyltransferase AR class, including 23S rRNA methyltransferase, which may confer resistance toward erythromycin and clindamycin, as demonstrated in a previous study [48] (Figure 1). In addition, we identified 500 genes predicted to belong to the sulfonamide-resistance class, which includes genes encoding enzymes counteracting sulfonamide antimicrobial agents, also known as *sul* genes [49] (Figure 1). The *sul* gene appears to be present as three variants in the investigated genomes, i.e., *sul1*, *sul2*, and *sul3*, all encoding a dihydropteroate synthase [49,50]. Interestingly, in the assessed genomes of the *Bifidobacterium* genus, the most prevalent gene variant was *sul3*, found in 90.4% of all identified sulfonamide-resistance genes.

Finally, the aminoglycoside class and the metronidazole class were the two least represented classes of *AR* genes in bifidobacteria, with just 73 and 64 identified genes predicted to be members of these two respective classes (Figure 1). Notably, *B. bifidum* AF11-25B was predicted to contain the highest number of genes encoding enzymes that counteract aminoglycoside antibiotics, such as streptomycin, kanamycin, and gentamicin.

Moreover, *B. bifidum* TF05-1 was the only strain whose chromosome contains a gene encoding a putative quinolone-resistance protein. This gene encodes a pentapeptide repeat protein, which is predicted to be involved in fluoroquinolone resistance [51,52,53].

Interestingly, comparing the identified *Bifidobacterium* resistome with AR determinants of other gut commensal, such as members of the *Lactobacillus* genus, we observed a lesser complexity of the resistome [54]. In fact, the *Lactobacillus* genus included different genes that could confer resistance toward a wide range of antibiotic molecules, such as vancomycin, erythromycin, and penicillin, but also tetracycline, chloramphenicol, and aminoglycoside antibiotics [55,56,57,58,59]. Furthermore, different *Escherichia coli* strains presented in their genomes *AR* genes that counteracted carbapenem antibiotics [60,61], whereas bifidobacteria seemed to be very sensitive to this antibiotic class, and their genomes do not encompass any genes that could confer resistance toward this antibiotic. Notably, a recent study based on metagenomics analyses of the human gut microbiota revealed that *Entrococcus* and *Enterobacter* genera possessed a very high antibiotic resistance load [62]. These genera presented *AR* genes that could counteract different antibiotics, such as trimethoprim/sulfamethoxazole, metronizadole, cycloserine, and cefixime [62]. Moreover, different studies have demonstrated the presence of *AR* genes in the genomes of the members of *Bacillus* genus, used as probiotic bacteria in functional food and for animal feed [54,63]. In the latter genus, macrolide-resistance genes have been identified on extra-chromosomal elements, tetracycline resistance genes, but also *cfr*-like genes (i.e., conferring resistance toward several classes of antibiotics, including phenicols, oxalozidinone, lincosamides, pleuromutilinis, and streptogramin) that have not been identified in the genomes of the members of the *Bifidobacterium* genus [64,65,66]. Our resistome analyses revealed a lack of specific *Bifidobacterium AR* genes, corroborating the safer behavior of the *Bifidobacterium* genus compared to other human gut commensals.

Although we do acknowledge the limitations of the in silico analysis in assigning antibiotic resistance functions to these identified genes, they are nonetheless considered to represent a potential arsenal to counter antimicrobial molecules.

### 3.2. The Predicted Mobilome of the Bifidobacterium Genus

The mobilome is defined as genetic elements that can move within a genome and between different genomes, including transposable elements, bacteriophages, and plasmids [21,22,23]. Similar to other members of the gut microbiota, it has been demonstrated that bifidobacteria possess genetic elements whose action is responsible for shaping their genomes [23,24,67,68]. In order to explore the mobile element repertoire of the *Bifidobacterium* genus, we analyzed the same 625 *Bifidobacterium* genomes as indicated above (Appendix A). Our analyses updated previously published data based on the reconstruction of 60 different bifidoprophage-like elements of 48 species of the *Bifidobacterium* genus.

A screening among the analyzed bifidobacterial genomes revealed 16,065 different genes encoding transposases, excluding genes that were truncated at the start or end codon (Appendix A). The sequence of each IS element was classified according to the ISFinder database [69], showing that members of the IS3 family are the most widespread among the *Bifidobacterium* genus (Appendix A). Notably, members of the *Bifidobacterium breve* species showed the highest number of IS elements, i.e., strains BR-06, BR-H29, BR-21, BR-L29, and BR-C29, ranging from 174 to 102 (Appendix A). Moreover, 16.5% of the analyzed genomes were predicted to contain less than 10 genes encoding transposases in their chromosomes while *Bifidobacterium commune* LMG 28292 does not appear to encompass any IS element at all (Appendix A).

Recently, Lugli et. al. recognized and classified all prophage-like elements (referred to as bifidoprophages) present in 48 genomes of type strains belonging to different bifidobacterial species [24] and Mavrich et al. characterized three of these identified groups of prophages integrated in members of *B. breve* and *B. longum* species by means of induction experiments [70]. In the current study, the screening for bifidoprophages was further extended to 625 different bifidobacterial genomes, resulting in the identification of 598 putative and apparently complete prophage sequences (Appendix A). Notably, the genomes of *Bifidobacterium biavatii* DSM 23969, *Bifidobacterium imperatoris* LMG 30297, and *Bifidobacterium cuniculi* LMG 10738 were predicted to contain the highest number of prophages in their genomes, i.e., seven, six, and five prophage-like elements, respectively (Appendix A). In order to evaluate the homology among the identified prophage-like elements, a genomic-based alignment clustering was performed. We observed the presence of four main homology clusters, in which the taxonomic origin of the corresponding *Bifidobacterium* hosts was highly heterogeneous. Each identified cluster showed several sub-clusters consisting of different prophage-like elements belonging to bifidobacterial strains of the same species, highlighting a sub-cluster phage specificity that appears to be host related (Figure 2). As reported in previous studies, prophages contribute to the genetic individuality of bacterial strains, containing many unique genes that in some instances may confer a fitness advantage to the host, such as a gene related to antibiotic resistance [71,72,73]. The mobile nature of phages may then allow the transfer of such advantageous genes to human/animal pathogens or to other non-resident microorganisms of the gut microbiota.

### 3.3. Identification of the Putative Mobile Resistome of the Bifidobacterium Genus

In order to evaluate the insurgence of *AR* genes located on or close to mobile elements, such as transposases and bifidoprophages, we investigated the flanking genes of the predicted resistome of the 625 bifidobacterial genomes. These regions may represent mobile genetic hotspots (MGHs) that promote HGT events, thereby transferring antibiotic resistance to other bacteria. We identified 201 putative MGHs distributed in 120 of the 625 *Bifidobacterium* strains studied. The number of *AR* genes involved in MGHs were very small compared to the total number of resistome genes (i.e., 13,870), representing less than 1.5% of the total *Bifidobacterium* resistome. Interestingly, we could not observe a correlation between a specific type of IS element and a class of *AR* genes.

As already noted in previous studies, 37 of the 41 strains of *Bifidobacterium animalis* subsp. *lactis* species contain a *tet*W gene flanked by a putative conjugative transposon (Figure 3) [74,75,76,77]. The *tet*W gene encodes a protein belonging to the Guanosine-5’-triphosphate (GTP)-binding elongation factor family that protects ribosomes from the translation inhibition activity of tetracycline [78]. Notably, this MGH is also present in 15 other genomes belonging to members of the *Bifidobacterium adolescentis*, *B. animalis* subsp. *animalis*, *B. breve*, *Bifidobacterium longum* spp., *Bifidobacterium pseudolongum* subsp. *pseudolongum*, and *Bifidobacterium pullorum* species. Remarkably, *tet*W appears to be well conserved among different species (Figure 3), suggesting the involvement of HGT events that could have transferred this tetracycline resistance gene to different bifidobacterial strains.

Interestingly, 67 MGHs involved prophage-like elements, which harbor a gene encoding for an UDP pyrophosphate phosphatase within their sequence (Figure 3), revealing a domain in the amino acid sequence that resembles a bacitracin-resistance protein (BacA) [79,80]. These 67 MGHs were present in members of three different *Bifidobacterium* species, i.e., *B. breve*, *B. longum* spp., and *Bifidobacterium pseudocatenulatum*, putatively conferring resistance to bacitracin through the phosphorylation of undecaprenol [79,80]. Prophages influence the biodiversity and abundance of bacteria in the human/animal intestinal tract, conferring new capabilities to their host [72]. The acquisition of a prophage-like element may thus confer a fitness advantage [72,73], in this particular case by conferring bacitracin resistance to these *Bifidobacterium* strains.

Moreover, a gene encoding a 23S rRNA methyltransferase flanked by a transposase was identified in 53 putative MGHs (Figure 3). In a recent study, Martinez et al. demonstrated the existence of a 23S rRNA methylase that confers erythromycin and clindamycin resistance to *B. breve* CECT7263 [48]. We found these MGHs in 10 different *Bifidobacterium* species, including *B. adolescentis*, *B. bifidum*, *B. breve*, *Bifidobacterium choerinum*, *Bifidobacterium kashiwanohense*, *B. longum* spp., *B. pseudocatenulatum*, *B. pseudolongum* subsp. *pseudolongum*, and *B. pullorum*. The highest occurrence of this genetic hotspot was in *B. breve* strains, where this hotspot was present in 15 out of 88 *B. breve* genomes analyzed. This methyltransferase is responsible for the enzymatic modification of the nucleotide sequence of the 23S rRNA gene, adding a methyl group, and preventing the linking of macrolide molecules [48]. Notably, the transposases that encompass these MGHs are predicted to be replicative transposons that may cause a rearrangement within bifidobacterial genomes, indicating that these MGHs rarely transfer to other genomes.

Remarkably, *B. longum* subsp. *longum* E18, isolated from healthy adult feces samples [81], is the only strain whose chromosome contains a prophage-like element, including a gene predicted to encode a protein with a complete beta-lactamase domain (Figure 3). Furthermore, the genome of strain *Bifidobacterium parmae* LMG 30295 contains a *van*Z homolog flanked by a predicted transposase-encoding gene, belonging to the transposon family IS256. The *van*Z gene is predicted to confer low-level resistance to the glycopeptide antibiotic, teicoplanin (Te), which prevents incorporation of D-alanine into peptidoglycan precursors [40]. This hotspot did not include a conjugative transposon, decreasing possible transfer events and bringing possible genomic rearrangements [82,83]. Therefore, more than 50% of putative MGHs identified encompassed transposons that cannot be classified as conjugative transposons, reducing possible HGT events involving *AR* genes, and corroborating previously published data [16].

The distribution of putative *AR* genes among analyzed bifidobacteria could be due to selective pressure imposed by intensive antibiotic use in their animal/human hosts, similar to what has been observed for lactic acid bacteria (LAB) [16,54]. These findings underline the safety of this genus and the very low frequency by which these *AR* genes may transfer to other members of the gut microbiota.

### 3.4. Assessment of Bacitracin A Resistance of Bifidobacterium spp

In order to validate our in silico predictions, we further investigated the antibiotic resistance of bifidobacterial strains whose genomes were shown to contain a *bac*A gene located in the sequence of a prophage-like element. Thus, in vitro measurements of MIC breakpoints for the bacitracin A antibiotic were monitored, including three *Bifidobacterium* strains, i.e., *B. breve* 1891B, *B. longum* subsp. *longum* 35B, and *B. longum* subsp. *infantis* ATCC 15697, whose genomes encompass a predicted *bac*A gene and three additional strains as a control, i.e., *B. breve* LMG 13208, *B. longum* subsp. *longum* LMG 13197, and *B. longum* subsp. *infantis* 1888B, whose chromosomes do not include a predicted *bac*A gene.

As indicated by this in silico analysis, those strains containing the *bac*A gene in their genomes exhibit a higher resistance level to bacitracin A (ranging from 16-fold to 32-fold) when compared to control strains (Appendix A). In this context, the bacitracin A breakpoints MIC values of *B. breve* 1891B, *B. longum* subsp. *longum* 35B, and *B. longum* subsp. *infantis* ATCC 15697 were, respectively, 16, 32, and 64 µg/mL, whereas the MIC values of the members of the control group were 2 µg/mL for *B. breve* LMG 13208 and 1 µg/mL for *B. longum* subsp. *longum* LMG 13197 and *B. longum* subsp. *infantis* 1888B (Appendix A). Statistical analyses were performed to corroborate the observed MIC differences, resulting in a significant growth difference between the two groups analyzed (*p*-value < 0.001) (Appendix A). These results confirmed the in silico-predicted resistance to bacitracin of those bifidobacterial strains possessing MGHs related to the *bac*A gene. The fact that the in silico analyses matched with the in vitro data highlighted the validity of an in silico resistome prediction [84].

## 4. Conclusions

Bifidobacteria are dominant members of the human/animal GIT, especially during the early stage of life. It has previously been demonstrated that the presence of this genus in the microbiota is associated with health-promoting effects [4]. In the current study, we reconstructed the resistome and the mobilome of members of the *Bifidobacterium* genus, evaluating genetic hotspots that could be involved in HGT events. The reconstructed putative resistome revealed that only a limited number of bifidobacterial genes are likely to be involved in putative AR spread. Moreover, the AR genetic arsenal of the *Bifidobacterium* genus seems to be less complex compared to the resistome of other gram-positive bacteria, such as members of the *Lactobacillus* genus, or other species included in food supplements and used as probiotics, such as members of the *Bacillus* genus [54,56,65,66,85,86]. Identified MGHs were restricted to less than 20% of the analyzed strains, of which most were isolated from the human GIT, suggesting the occurrence of AR in members of the human microbiota because of intense antibiotic therapies. Remarkably, the acquisition of phages encompassing *AR* genes in their sequence could confer ecological advantages, increasing the biological fitness of their host [72,73]. Nevertheless, the vast majority of identified MGHs in the *Bifidobacterium* genus are unlikely to be transferred to other microorganisms, due to the transposition mechanisms of the identified IS elements flanking putative *AR* genes. Moreover, in vitro bacitracin A antibiotic resistance tests based on bifidobacterial strains containing *bac*A located in an MGH confirmed our in silico prediction. Finally, these findings underpin the safety of the *Bifidobacterium* genus compared to other taxa, such as *Escherichia coli* and members of the Gammaproteobacteria class, which were shown to contribute to a high antibiotic resistance load in the human microbiota [87].

## Figures and Tables

**Figure 1 microorganisms-07-00638-f001:**
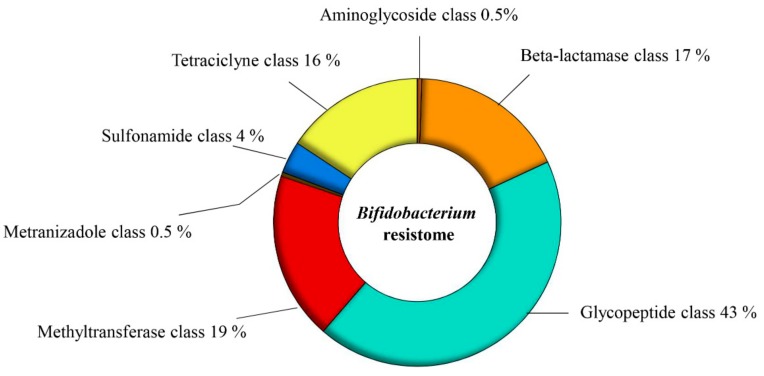
Predicted resistome of the *Bifidobacterium* genus. Abundance of different predicted antibiotic resistance gene classes identified among the 625 analyzed *Bifidobacterium* genomes.

**Figure 2 microorganisms-07-00638-f002:**
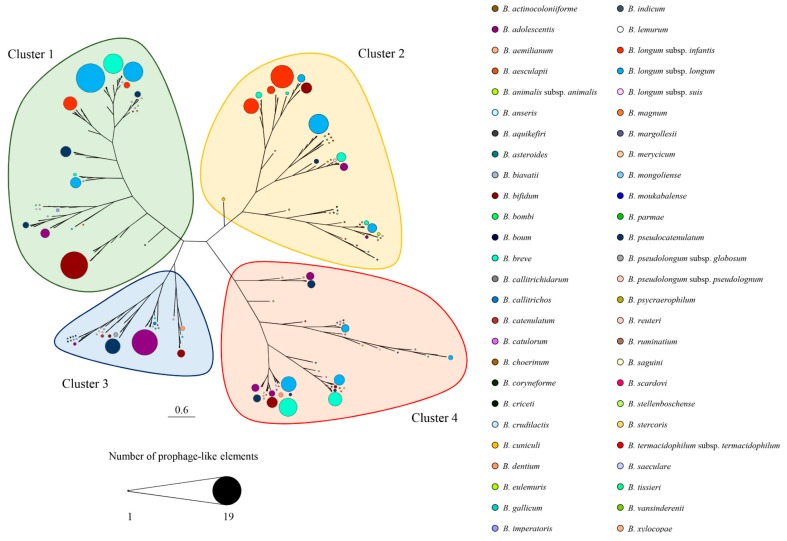
Phylogenetic tree of identified bifidoprophages. Genomic alignment-based clustering of 598 prophages identified within bifidobacterial strain genomes. Each colored dot represents the *Bifidobacterium* host species origin of a given bifidoprophage. The dot size refers to the number of prophage-like sequences identified within the same branch tree. The four different clusters are highlighted with different colors.

**Figure 3 microorganisms-07-00638-f003:**
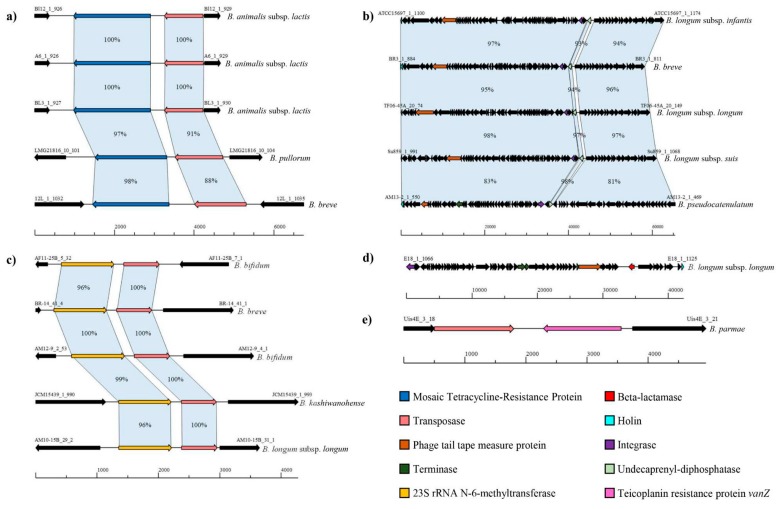
Mobile genetic hotspots identified in the *Bifidobacterium* genus. Bifidobacterial genomic regions containing putative Mobile Genetic Hotspots (MGHs). Different species and gene names are reported next to each genomic region. Panels (**a**–**c**) show the genomic regions conserved among different *Bifidobacterium* species. Panels (**d**,**e**) display unique mobile genetic hotspots identified in *B. longum* subsp. *longum* E18 and *B. parmae* LMG 30,295 strains. Each arrow indicates a gene and the different colors indicate the function of the gene product.

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
