# Peer review of "Mobilome and Resistome Reconstruction from Genomes Belonging to Members of the Bifidobacterium Genus"

_microorganisms, 2019, doi:10.3390/microorganisms7120638_

Round 1

Reviewer 1 Report

The manuscript by Mancino et al. describes an extensive in silico analysis of Bifidobacterium spp. resistome. Members of Bifidobacterium genus are crucial players in establishment of intestine microbial community during the neonatal period. Despite many studies of in silico resistome of metagenomic samples have been published so far, work of Mancino et al. looks very advantageous due to its focus on specific group of microorganisms. Knowledge on the distribution of antibiotic-resistant genes in Bifidobacterium might have a significant contribution for the further development of microbiota-correction techniques and treatments. Nevertheless, it is a bit frustrating, that authors have not compared the resistome of Bifidobacteria with the resistomes of other major players of the microbial consortium of human gut (Lactobacillus, Faecalibacterium etc). If any Bifidobacterium specific ARG or mobile elements were found, that would significantly improve the scientific soundness.

Also, there are some minor corrections that would improve the quality of the presentation of obtained results:

L.22-23  Due to the issues of assembling genomic repeats, mobile genetic elements are often located at the ends of contigs, and therefore the analysis of their abundance and distribution might be compromised using draft and low-quality genome assemblies. Please, briefly specify in the abstract the type of the data you have used.

L.69-74  Please, describe, how did you perform the quality check of genome assemblies deposited in NCBI. If no QA was performed, then, please include these steps into analysis (for example, CheckM tool for the estimation of assembly completeness and contamination)

L.77-78  Please describe in more detail, how the custom database was created. What were the criteria of sequence selection? Was the sequence clustering performed? What was the number of sequences in the final database?

L.84-89  Again, please describe the custom database, or perform an additional phage search with another tool (http://phaster.ca/) to confirm your results.

L.93-94  1e-18 ?  Why this value?

L.109 FIGTREE is a tree viewer and editor. Please specify, how the tree was constructed, not visualized.

L.172-206 (section 3.2)  Please, include a figure or table, describing the distribution of IS element families. Did you see any correlation between specific types of IS elements and classes of AR genes?

Author Response

Enclosed please find the submission of the revised manuscript, entitled “Mobilome and resistome reconstruction from genomes belonging to members of the Bifidobacterium genus” for consideration of publication in Microorganisms.

We have addressed all comments and suggestions made by the expert referees and have implemented all the proposed changes. All comments were considered very helpful and valuable, significantly improving the manuscript in its overall quality.

In the following, we discuss the individual suggestions/comments made by the reviewer 1 in detail:

Reviewer 1:

As suggested by this referee, we have added information about the resistome of other major microbial players of the human gut (Lines 171-190). In accordance with this reviewer, we have introduced in the abstract section additional details about which genomes were used for our analyses (Lines 22-23). As suggested by this referee, we added details about the quality check performed among bifidobacterial genome assemblies (Lines 70-72). As indicated by this reviewer, we have described in detail how the IS elements database was developed. We have added the total number of sequences presented in the final database (Lines 78-79). In accordance with this referee, we have described the bifidoprophage custom database employed for the analysis, i.e., a validated database composed by prophage sequences previously collected (Lugli et al., Environ Microbiol, 2016), and the criteria for the sequence selection (Lines 88-90). As suggested by this reviewer, we have explained the cutoff choice for the resistome analysis (Lines 98-99). As indicated by this referee, we have specified how we constructed the clustering tree of bifidoprophages (Lines 111-116).

8. As suggested by this reviewer, we included a supplementary table (table S2) describing the distribution of IS family in the Bifidobacterium genus. We could not observe correlations between a specific type of IS elements and classes of AR genes (Line 210, Lines 248-249 and Table S2).

Reviewer 2 Report

In the present paper the authors analyzed public available genome data base of as many as 625 bifidobacterial strains belonging to 67 (sub)species, focusing on the antibiotics resistant genes (resistsome) and mobile genetic elements (mobilome). They also tried to show the genetic insertion hot spots.

Comments.

1. Since the genus Bifidobacteriu are among important intestinal habitats associated with human and other host health, the paper will provide useful knowledge from 625 strains how the antibiotic genes and movable elements are distributed in these organisms.

2.  The related papers (Ref, 16, 23, 24, 43, 44) have been published by the same author group, and the term of reconstruction is used in title, please try to highlight the findings in the present paper in comparison with the previous works.

3. The data on mobilosome and resistosome in the Bifidobacterium obtained in this study are also true for other Gram-positive lactic acid bacteria and Gram-negative ones such as Bacteroides and E. coli. Please discuss this point if the data are available.

4. Fig. 1 results reflect the dosage of antibiotics?

5. It would be more useful if the authors could analyze the mobilosome and resistosome data based on the sources of Bifidobacterium strains (for instance, the ones from infant/adult guts, different countries etc.)

6. Fig. 3. Explain in more detail. Difficult to understand this figure from this Fig. legend.

7. Lines 216 – 218, OK? How less than 1.5% is obtained?

Author Response

Enclosed please find the submission of the revised manuscript, entitled “Mobilome and resistome reconstruction from genomes belonging to members of the Bifidobacterium genus” for consideration of publication in Microorganisms.

We have addressed all comments and suggestions made by the expert referees and have implemented all the proposed changes. All comments were considered very helpful and valuable, significantly improving the manuscript in its overall quality.

In the following, we discuss the individual suggestions/comments made by the reviewer 2 in detail:

Reviewer 2:

We agree with this reviewer. As mentioned in this comment, our manuscript analyzed antibiotic resistance genes and mobile genetic elements of Bifidobacterium genus, which is considered an important gut commensal. As suggested by this referee, we have highlighted the main findings of this study in respect to the previously published works (Lines 130-132 and Lines 204-206). As indicated by this reviewer, we have discussed the comparison between the resistome of the genus Bifidobacterium and that of other suggested taxa (Lines 171-190). Figure 1 depicts the distribution of AR classes identified in the “in silico” analysis of the Bifidobacterium resistome. These results did not reflect the antibiotics dosage. We would like to thank the referee for this comment. In fact, the correlation between the mobilome and resistome data with the ecological origin of the bifidobacterial strains could be very interesting. We tried to collect this data, but we noticed that ecological information is not available for all strains analyzed. In accordance with this referee, we have explained in more detail Fig. 3 (Lines 296-299). As suggested by this reviewer, we specified how the 1.5 % of the identified resistome was evaluated (Lines 246-248).